# Fractional Modeling of Viscous Fluid over a Moveable Inclined Plate Subject to Exponential Heating with Singular and Non-Singular Kernels

**Aziz Ur Rehman** [1], **Muhammad Bilal Riaz** [1,2,*] , **Wajeeha Rehman** [1], **Jan Awrejcewicz** [2] and **Dumitru Baleanu** [3,4,*]

[1] Department of Mathematics, University of Management and Technology, Lahore 54770, Pakistan; s2019265005@umt.edu.pk (A.U.R.); ar4059481@gmail.com (W.R.)
[2] Department of Automation, Biomechanics and Mechatronics, Lodz University of Technology, 1/15 Stefanowskiego Str., 90-924 Lodz, Poland; jan.awrejcewicz@p.lodz.pl
[3] Department of Mathematics, Cankaya University, Ankara 06530, Turkey
[4] Institute of Space Sciences, 06530 Bucharest, Romania
[*] Correspondence: Muhammad.riaz@p.loz.pl (M.B.R.); dumitru.baleanu@gmail.com (D.B.)

**Abstract:** In this paper, a new approach to investigating the unsteady natural convection flow of viscous fluid over a moveable inclined plate with exponential heating is carried out. The mathematical modeling is based on fractional treatment of the governing equation subject to the temperature, velocity and concentration field. Innovative definitions of time fractional operators with singular and non-singular kernels have been working on the developed constitutive mass, energy and momentum equations. The fractionalized analytical solutions based on special functions are obtained by using Laplace transform method to tackle the non-dimensional partial differential equations for velocity, mass and energy. Our results propose that by increasing the value of the Schimdth number and Prandtl number the concentration and temperature profiles decreased, respectively. The presence of a Prandtl number increases the thermal conductivity and reflects the control of thickness of momentum. The experimental results for flow features are shown in graphs over a limited period of time for various parameters. Furthermore, some special cases for the movement of the plate are also studied and results are demonstrated graphically via Mathcad-15 software.

**Keywords:** Laplace transform; viscous fluid; ramped conditions; system parameters; porous material

## 1. Introduction

Fluids which are electrically conducted magneto-hydrodynamics (MHD) have wide applications in chemical engineering, modern technology and geophysical environments [1,2], but double diffusive convection is a mixing process due to the interaction of different components of fluid having different density gradients and rates of diffusion [3]. Oceanography is the simplest example of this phenomena, in which the concentration of salt and heat exists with distinct gradients and they diffuse with different rates. For more details we refer to [4]. Natural or free convection flow which occurs in the presence of temperature gradient is one of the most significant modes used to transfer heat and mass in many geophysical phenomena and modern technological fields. At present, most researchers are interested in focusing their attention on the dynamical systems which have rich applications concerning magnetic fields. Moreover, over the years, the study of mass and thermal transport phenomenon for magneto-hydrodynamic (MHD) natural convective flow of fluids under the impact of electrical conduction has gained much popularity in view of their applications in meteorology, power elevators, chemical engineering, aerodynamic heating, geophysics, purification of mineral oil and solar physics. Results for this type of motion for the case of viscous fluids over vertical planes are developed for

diversified boundary conditions, for example, for an impulsively moving plate with radiation effects and ramp wall temperature [5], dynamics of a fluid of heat absorbing type with mass transfer [6], analytical study with a random unsteady shear stress boundary condition [7]. The MHD free convective flow passing through a micro-channel along with the conditions of temperature jump and velocity slip on the boundary has been studied in [8]. For other related and useful investigations, see [9–12] and the references therein. In all these investigations, it is remarkable to confer that the uniform magnetic flux in the fluid is unyielding. Narahari and Debnath [13] conducted a useful analysis of MHD-natural convective-flow by way of fixed heat flux with two cases; namely, either the intensity of magnetic field is fixed respective to the fluid (MFFRF) or the area around the magnet is fixed respective to the plate (MFFRP). Later, the results are obtained by Shah et al. [14,15] for MHD natural convective flow up an erected plate through chemical reaction with varying temperature of the plate in the cases for MFFRF and MFFRP. Furthermore, instability of free convection and transition to turbulence on inclined plates have great significance due to the fact that they are associated with cryogenic tanks and thermal stratification in heat exchanges, etc. Among many investigations with dynamical applications, existing in the literature, regarding natural flow of convective fluid through the inclined plates in porous media, Sparrow and Hussar [16] studied the natural convection flow on inclined plates coupled with generation of longitudinal vertices. Some notable investigations that have been carried out for free convective flow of viscous fluid on inclined plates can be seen in the references [17,18].

Nowadays, fractional order calculus—the branch of mathematics—has been rising vastly due to its exclusive significant features in science and engineering that are absent in non-fractional calculus, which deals with an arbitrary order of integration and differentiation. Fractional differential equations are massively applied to model various daily life physical problems because fractional calculus has memory effects, such as problems in fluid flow, diffusion, relaxation, reaction–diffusion relaxation, oscillation, dynamical processes, retardation processes in complex systems and many more engineering processes. For this reason, classical models are unable to anticipate the preceding state of the processes. In literature, most of the studies are focused on flow problems relative to several fractional operators with local and non-local kernels such as Caputo, Atangana–Baleanu, Caputo–Fabrizio, and a few others [19–22]; they indicate the current state but also the future state of a system. Riaz et al. [23] investigated the effect of ramped conditions on temperature and velocity by considering fractionalized convective flow model. Moreover, there is a comparative study for fractional model of MHD Maxwell fluid to anticipate the heat effect by Riaz et al. [24]. Some other fractional associated studies are discussed in detail; see for instance [25,26]; most of the studies are focused on flow problems with non-integer differential operators, heat transport MHD Jeffrey fluid movement and second grade fluid.

In this communication, the purpose of this exploration is to investigate the general study of double diffusive magneto-free-convection flow for viscous fluid presented in non-dimensional form, and to analyze the general motions of the oscillating inclined plate constituted in a porous material, with the existence of an oblique externally electromagnetic field whether moving or fixed, consistent with the porous layered plate. The thermal transport phenomenon is discussed in the presence of constant concentration coupled with thermal conductivity and first order chemical reaction. The influence of different angles is discussed on the fluid velocity that the plate makes with the vertical and oblique angles that the magnetic lines make with the porous layered inclined plate. Moreover, the fluid velocity expressed in terms of thermal, mechanical, concentration contributions, and solution expressions for such components are obtained. In the proposed model, the temperature distribution is presented for the general function of time, and general motions of the plate are considered. Special cases for the movement of the plate are studied and several results are recovered corresponding to viscous fluids exist in the literature as limiting cases by assigning different functions and parametric values in the general solutions of the problem. The solutions relating to swaying movements of the plate are

also discussed and demonstrated, those being the sum of transient and parts and steady state. Furthermore, the consequences of different related physical parameters, such as chemical reaction parameter $R_c$, magnetic field $M$, dimensionless time $\eta$, Schmidt number $Sc$, Ratio of the buoyancy forces $N$ and effective Prandtl number $Pr_{eff}$, on non-dimensional velocity, concentration and energy are discussed in detail and demonstrated graphically using Mathcad-15 software. Motivated by the above consideration, the main theme of this manuscript is to have the significance of fractional models corresponding to ordinary existing models for second grade fluid. Further, some results are recovered from the existing literature as limiting cases to validate our obtained results.

## 2. Mathematical Model

We consider the time dependent, incompressible, electrically conducting natural convective movement of viscous fluid over a porous inclined plate, which is also non-conductive having infinite length. The x-axis is assumed along the vertical, and $v$ takes an angle that the plate makes with the vertical, and this angle lies between 0 and $\frac{\pi}{2}$, i.e., $0 \leq v \leq \frac{\pi}{2}$. An unvarying magnetic intensity with strength $\overrightarrow{B_0} = (B_0 \cos \hbar, B_0 \sin \hbar)$ is exerted, where $\hbar$ is a slanted angle that magnetic lines make with the porous layered inclined plate, along with the assumptions asserted that the magnetic intensity is supposed to be fixed for the plate or the fluid. Initially, it is supposed that both fluid and plate are static with fixed species concentration $C_\infty$ and temperature $T_\infty$. As time $\eta = 0^+$, motion in the plate starts that excels the gravitational pull with certain $\zeta_0 f(\eta)$ in opposition. Moreover, the temperature is stabilized at the expense of relation in the form $T_\infty + T_w\left(1 - ae^{-b\eta}\right)$, whereas concentration is sustained at $C_w$. Here, $f(\cdot)$ is a continuous piece-wise function that dies out at $\eta = 0$; $\zeta_0$ is assumed to be a constant with dimension of velocity. The choice of an appropriate Cartesian coordinate system and use the Boussinesq's approximation [27,28] leads the relevant problem statement into the following system of governing equations:

$$\frac{\partial \zeta(\psi,\eta)}{\partial \eta} = v\frac{\partial^2 \zeta(\psi,\eta)}{\partial \psi^2} + g\beta_T(T(\psi,\eta) - T_\infty)\cos(v)$$

$$+ g\beta_C(C(\psi,\eta) - C_\infty)\cos(v) - \left[\frac{\sigma B_0^2 \sin^2(\hbar)}{\rho} + \frac{v}{k}\right]\zeta(\psi,\eta), \tag{1}$$

$$\rho C_p \frac{\partial T(\psi,\eta)}{\partial \eta} = k\frac{\partial^2 T(\psi,\eta)}{\partial \psi^2} - \frac{\partial q_r}{\partial \psi}, \tag{2}$$

$$\frac{\partial C(\psi,\eta)}{\partial \eta} = \delta_m \frac{\partial^2 C(\psi,\eta)}{\partial \psi^2} - R_c(C(\psi,\eta) - C_\infty). \tag{3}$$

with initial and boundary conditions as defined below:

$$\zeta(\psi,0) = 0, \quad T(\psi,0) = T_\infty, \quad C(\psi,0) = C_\infty, \quad \frac{\partial \zeta(\psi,0)}{\partial \eta} = 0, \quad \psi \geq 0, \tag{4}$$

$$\zeta(0,\eta) = \zeta_0 H(\eta)f(\eta), \quad T(0,\eta) = T_\infty + T_w(1 - ae^{-b\eta}), \quad C(0,\eta) = C_w \quad \eta > 0, \zeta_0 \neq 0. \tag{5}$$

$$\zeta(\psi,\eta) \to 0, \quad T(\psi,\eta) \to \infty, \quad C(\psi,\eta) \to \infty \text{ as } \psi \to \infty. \tag{6}$$

where $\zeta(\psi,\eta), T(\psi,\eta), C(\psi,\eta), \rho, \beta_T, \beta_C, v, k, q_r, g$ and $C_p$ denote the fluid velocity, temperature of the fluid, concentration, density, coefficient of volumetric thermal expansion, coefficient of volumetric expansion for concentration, kinematic viscosity, thermal conductivity, radiative heat flux, gravitational acceleration and heat capacity at constant pressure,

respectively. The Rosseland diffusion approximation is adapted (for an optically thick fluid) [11,29]

$$q_r = -\frac{4}{3}\frac{\sigma_1}{k_R}\frac{\partial T^4(\psi,\eta)}{\partial \psi}, \tag{7}$$

where $k_R$ denotes the Rosseland mean attenuation coefficient and $\sigma_1$ represents the Stefan–Boltzman constant. For the case, the disparity between T (fluid temperature) and $T_\infty$ (free stream temperature) is too small, i.e., $|T - T_\infty| << 0$.

The set of dimensionless quantities are introduced:

$$\zeta^* = \frac{\zeta}{\zeta_0}, \quad \psi^* = \frac{\zeta_0}{\nu}\psi, \quad \theta = \frac{T - T_\infty}{T_w}, \quad N = \frac{\beta_C(C_w - C_\infty)}{\beta_T T_w}, \quad C^* = \frac{C - C_\infty}{C_w - C_\infty},$$

$$M = \frac{\sigma_0\beta_0^2\nu}{\rho\zeta_0^2}, \quad \eta^* = \frac{\zeta_0^2}{\nu}\eta, \quad N_r = \frac{16}{3}\frac{\sigma_1}{kk_R}T_\infty^3, \quad Pr = \frac{\mu C_p}{k}, \quad \nu = \eta_0\zeta_0^2, \quad Sc = \frac{\nu}{\delta_m},$$

$$f^*(\eta^*) = f(\frac{\nu}{\zeta_0^2}\eta^*), \quad Pr_{eff} = \frac{Pr}{1 + N_r}, \quad \frac{1}{K} = \frac{\nu^2\phi}{k\zeta_0^2}, \quad b^* = \frac{\nu}{\zeta_0^2}b, \quad R_c = \frac{\nu}{\zeta_0^2}R_c.$$

After employing the dimensionless quantities, ignore the asterisk $*$ notation, the following partial differential equations in dimentionless form are derived as:

$$\frac{\partial \zeta(\psi,\eta)}{\partial \eta} = \frac{\partial^2 \zeta(\psi,\eta)}{\partial \psi^2} + \theta(\psi,\eta)\cos(\nu) + NC(\psi,\eta)\cos(\nu) - K\zeta(\psi,\eta) - M\sin^2(\hbar)\zeta(\psi,\eta), \tag{8}$$

$$\frac{\partial \theta(\psi,\eta)}{\partial \eta} = \frac{1}{Pr_{eff}}\frac{\partial^2 \theta(\psi,\eta)}{\partial \psi^2}, \tag{9}$$

$$\frac{\partial C(\psi,\eta)}{\partial \eta} = \frac{1}{Sc}\frac{\partial^2 C(\psi,\eta)}{\partial \psi^2} - R_c C(\psi,\eta). \tag{10}$$

with conditions in dimensionless form as:

$$\zeta(\psi,0) = 0, \quad \theta(\psi,0) = 0, \quad C(\psi,0) = 0, \tag{11}$$

$$\zeta(0,\eta) = f(\eta), \quad \theta(0,\eta) = 1 - ae^{-b\eta}, \quad C(0,\eta) = 1 \quad \eta > 0. \tag{12}$$

$$\zeta(\psi,\eta) \to 0, \quad \theta(\psi,\eta) \to 0, \quad C(\psi,\eta) \to 0 \text{ as } \psi \to \infty. \tag{13}$$

## 3. Mathematical Preliminaries

In this section, we shall discuss the basic definitions with properties of some significant and applicable special functions.

*Special Functions*

As stated in the introduction, for getting the analytical solution from a differential equation with the application of Laplace integral transformation, the Laplace inverse transformation of some terms in the differential equation is not trivial. To tackle this problem, the solutions are expressed in the form of some special functions [30–32]. A number of special functions are listed in the literature and some are mentioned here, for instance, Mittag-Leffler [33], Robotnov and Hartley's [34], Lorenzo and Hartley's [35] Generalized G and R functions, etc. Such functions establish the results in precise form and play an important role in interpreting the linear integer order differential equation corresponding to IVPs.

In the following, we define the above mentioned special function definitions along with their Laplace transformations and some special cases as:

1. **Mittag-Leffler function.** The Mittag-Leffler function is the generalization of the exponential function and is defined as [33]

$$E_\imath(t) = \sum_{\wp=0}^{\infty} \frac{t^\wp}{\Gamma(\wp\imath+1)} \; ; \imath > 0.$$

The exponential function is a special case of this function; for $\imath = 1$, we get

$$E_1(t) = \sum_{\wp=0}^{\infty} \frac{t^\wp}{\Gamma(\wp+1)} = e^t.$$

Moreover,

$$\mathcal{L}\left\{E_\imath\left(-at^\imath\right)\right\} = L\left\{\sum_{\wp=0}^{\infty} \frac{(-a)^\wp t^{\wp\imath}}{\Gamma(\wp\imath+1)}\right\} = \frac{q^\imath}{q(q^\imath+a)}; \; \imath > 0.$$

2. **Erdelyi's function.** This function is the generalization of the Mittag-Leffler function and is described as [36]

$$E_{\imath,\beta}(t) = \sum_{\wp=0}^{\infty} \frac{t^\wp}{\Gamma(\wp\imath+\beta)} \; ; \imath, \beta > 0.$$

Setting $\beta = 1$, we have

$$E_{\imath,1}(t) = \sum_{\wp=0}^{\infty} \frac{t^\wp}{\Gamma(\wp\imath+1)} = E_\imath(t).$$

For $\imath = 1$ and $\beta = 2$, we have

$$E_{1,2}(t) = \frac{e^t - 1}{t}.$$

Similarly, for $\imath = \frac{1}{2}$ and $\beta = 1$, we get

$$E_{\frac{1}{2},1}(t) = e^{t^2} erfc(-t)$$

When $\imath = 2$ and $\beta = 2$, we have

$$E_{2,2}\left(t^2\right) = \frac{\sinh(t)}{t},$$

where $erfc(t) = \frac{2}{\sqrt{\pi}} \int_t^\infty e^{-u^2} du$ [30] is known as the complementary error function. Further,

$$\mathcal{L}\left\{E_{\imath,\beta}(t)\right\} = \sum_{\wp=0}^{\infty} \frac{\Gamma(\wp+1)}{\Gamma(\wp\imath+\beta)} \frac{1}{q^{\wp+1}}; \; \imath, \beta > 0.$$

3. **Robotnov and Hartley function.** This was presented by Hartley and Lorenzo [34] and later on studied by Robotnov for utilization in solid mechanics as well. It is confined as

$$F_\imath(-at) = t^{\imath-1} \sum_{\wp=0}^{\infty} \frac{(-a)^\wp t^{\wp\imath}}{\Gamma(\wp\imath+1)} \; ; \imath > 0.$$

Here,

$$E_\imath\left(-at^\imath\right) = \sum_{\wp=0}^{\infty} \frac{(-a)^\wp t^{\wp\imath}}{\Gamma(\wp\imath+1)}, F\imath(-at) = t^{\imath-1}E_\imath\left(-at^\imath\right),$$

so

$$\mathcal{L}\{F_\imath(-at)\} = \frac{1}{q^\imath + a}; \ \imath > 0.$$

4. **Miller and Ross' function.** It was proposed by Miller, and Ross [37]. This function is stated as:

$$E_t(\imath, a) = t^\imath \sum_{\wp=0}^\infty \frac{(at)^\wp}{\Gamma(\imath + \wp + 1)} \ ; \ Re(\imath) > 1,$$

$$\mathcal{L}\{E_t(\imath, a)\} = \frac{q^{-\imath}}{q - a}; \ Re(\imath) > 1.$$

5. **Generalized R-function.** Lorenzo and Hartley [35] developed this function; it is written as:

$$R_{\imath,\beta}(a, t) = \sum_{\wp=0}^\infty \frac{a^k t^{(\wp+1)\imath - \beta - 1}}{\Gamma((\wp+1)\imath - \beta)} \ ; \ Re(\imath - \beta) > 0.$$

It is easy to see that $R_{1,0}(a, t) = e^{at}$, $aR_{2,0}(-a^2, t) = \sin(at)$ and $R_{2,1}(-a^2, t) = \cos(at)$. When $a = 1, \beta = \imath - 1$, we get

$$R_{\imath,\imath-1}(1, t) = \sum_{\wp=0}^\infty \frac{(t^\imath)^\wp}{\Gamma(\wp\imath + 1)} = E_\imath\left(t^\imath\right).$$

Similarly, for $a = 1, \beta = \imath - v$, yields

$$R_{\imath,\imath-v}(1, t) = t^{v-1} \sum_{\wp=0}^\infty \frac{(t^\imath)^\wp}{\Gamma(\wp\imath + v)} = t^{v-1} E_{\imath,v}\left(t^\imath\right).$$

Moreover,

$$\mathcal{L}\{R_{\imath,\beta}(a, t)\} = \frac{q^\beta}{q^\imath - a}; \ Re(\imath - \beta) > 0, \ Re(q) > 0.$$

6. **Generalized G-function.** Lorenzo and Hartley [35] also introduced this function which is the generalization of *R*-function and is specified as:

$$G_{\imath,b,j}(a, t) = \sum_{\wp=0}^\infty \frac{a^\wp \Gamma(j + \wp)}{\Gamma(j)\Gamma(\wp + 1)} \frac{t^{(j+\wp)\imath - b - 1}}{\Gamma((\wp+1)\imath - b)} \ ; \ Re(\imath j - b) > 0.$$

For $j = 1$, we have

$$G_{\imath,b,1}(a, t) = \sum_{\wp=0}^\infty \frac{a^\wp t^{(1+\wp)\imath - b - 1}}{\Gamma((\wp+1)\imath - b)} = R_{\imath,b}(a, t).$$

Moreover,

$$\int_0^s G_{\imath,b,j}(a, t)dt = \sum_{\wp=0}^\infty \frac{a^\wp \Gamma(j + \wp)}{\Gamma(j)\Gamma(\wp + 1)} \frac{s^{(j+\wp)\imath - b}}{((j+\wp)\imath - b)\Gamma((\wp+1)\imath - b)}$$

$$= \sum_{\wp=0}^\infty \frac{a^\wp \Gamma(j + \wp)}{\Gamma(j)\Gamma(\wp + 1)} \frac{s^{(j+\wp)\imath - b}}{\Gamma((\wp+1)\imath - b + 1)} = G_{\imath,b-1,j}(a, s).$$

Moreover,

$$\mathcal{L}\left\{G_{\imath,b,j}(a, t)\right\} = \frac{q^b}{(q^\imath - a)^j}; \ Re(\imath j - b) > 0, \ Re(q) > 0, \ \left|\frac{a}{q^\imath}\right| < 1.$$

Next, we define Caputo, CF and ABC fractional operators used in this paper to fractionalize the proposed problem.

- Caputo fractional operator having power law kernel is described as:

$$^{C}D_{\eta}^{\wp}f(z,\eta) = \frac{1}{\Gamma(1-\wp)} \int_{0}^{\eta} \frac{1}{(\eta-\tau)^{\wp}} \frac{\partial f(z,\tau)}{\partial \tau} d\tau, \qquad 0 < \wp < 1.$$

with Laplace transformation

$$\mathcal{L}\left(^{C}D_{\eta}^{\wp}f(z,\eta)\right) = s^{\wp}\mathcal{L}(f(z,\eta)) - s^{\wp-1}f(z,0).$$

- CF fractional operator with a non-singularized and local kernel is described as:

$$^{CF}D_{\eta}^{\wp}f(z,\eta) = \frac{1}{1-\wp} \int_{0}^{\eta} exp\left(-\frac{\wp(\eta-\tau)}{1-\wp}\right) \frac{\partial f(z,\tau)}{\partial \tau} d\tau, \quad 0 < \wp < 1.$$

Its Laplace transformation is obtained as:

$$\mathcal{L}\left(^{CF}D_{\eta}^{\wp}f(z,\eta)\right) = \frac{s\mathcal{L}(f(z,\eta)) - f(z,0)}{(1-\wp)s + \wp}.$$

- The Atangana–Baleanu fractional operator in a Caputo sense (ABC) with non-singularized and non-local kernel is defined in the following way:

$$^{ABC}D_{\eta}^{\wp}f(z,\eta) = \frac{1}{1-\wp} \int_{0}^{\eta} E_{\wp}\left(-\frac{\wp(\eta-\tau)^{\wp}}{1-\wp}\right) \frac{\partial f(z,\tau)}{\partial \tau} d\tau, \quad 0 < \wp < 1.$$

Its Laplace transformation is obtained as:

$$\mathcal{L}\left(^{ABC}D_{\eta}^{\wp}f(z,\eta)\right) = \frac{s^{\wp}\mathcal{L}(f(z,\eta)) - s^{\wp-1}f(z,0)}{(1-\wp)s^{\wp} + \wp}.$$

where $\wp$ is named as the fractional parameter.

## 4. Solution of the Problem

To get the solution of the considered problem, it is extremely essential to point out that without finding the expressions for the temperature and concentration we cannot established the expression of fluid velocity that is our target. The exact expressions for the temperature and concentration are to be found using the modern definition of CF and ABC non-integer operators from Equations (9) and (10).

### 4.1. Exact Solution of Heat Profile with CF Time Fractional Derivative

The fractional model for Equation (9) formulated on the base of the Caputo–Fabrizio fractional time derivative is provided as:

$$\frac{d^{2}\theta(\psi,\eta)}{d\psi^{2}} - Pr_{eff}{}^{CF}D_{\eta}^{\alpha}\theta(\psi,\eta) = 0. \tag{14}$$

Applying the Laplace Transformation technique to write the solution of Equation (14) with conditions as Equations (11)–(13), we have

$$\frac{d^{2}\bar{\theta}(\psi,q)}{d\psi^{2}} - Pr_{eff}\frac{q}{(1-\alpha)q + \alpha}\bar{\theta}(\psi,q) = 0. \tag{15}$$

with

$$\bar{\theta}(0,q) = \frac{1}{q} - \frac{a}{q+b} \quad \text{and} \quad \bar{\theta}(\psi,q) \to 0 \text{ as } \psi \to \infty. \tag{16}$$

and its solution is given by

$$\bar{\theta}(\psi,q) = \chi_1 e^{\psi \sqrt{\frac{qP_{eff}}{(1-\alpha)q+\alpha}}} + \chi_2 e^{-\psi \sqrt{\frac{qP_{eff}}{(1-\alpha)q+\alpha}}}. \tag{17}$$

we applied conditions for temperature given by Equation (16) to determined unknown constants $\chi_1$ and $\chi_2$; we get

$$\bar{\theta}(\psi,q) = \left(\frac{1}{q} - \frac{a}{q+b}\right)e^{-\psi \sqrt{\frac{qP_{eff}}{(1-\alpha)q+\alpha}}}. \tag{18}$$

which can be expressed as

$$\bar{\theta}(\psi,q) = \bar{\theta}_1(\psi,q) - a\bar{\theta}_2(\psi,q). \tag{19}$$

To get the required solution of Equation (19), Laplace inverse transformation is used, which is written as:

$$\theta(\psi,\eta) = \theta_1(\psi,\eta) - a\theta_2(\psi,\eta). \tag{20}$$

where

$$\theta_1(\psi,\eta) = \mathcal{L}^{-1}\left\{\frac{e^{-\psi\sqrt{\frac{qP_{eff}}{(1-\alpha)q+\alpha}}}}{q}\right\}$$

$$= 1 - \frac{2P_{eff}}{\pi}\int_0^\infty \frac{Sin(\frac{\psi}{\sqrt{1-\alpha}}x)}{x(P_{eff}+x^2)}e^{\left(-\frac{\alpha}{1-\alpha}\eta x^2\right)}dx,$$

$$\theta_2(\psi,\eta) = (\theta_3 * \theta_4)(\eta),$$

$$\theta_3(\psi,\eta) = \mathcal{L}^{-1}\left\{\frac{1}{q+b}\right\} = e^{-b\eta},$$

$$\bar{\theta}_4(\psi,q) = e^{-\psi\sqrt{\frac{qP_{eff}}{(1-\alpha)q+\alpha}}}, \tag{21}$$

It is difficult to find $\theta_4(\psi,\eta)$ from exponential form, so we express $\bar{\theta}_4(\psi,q)$ in its equivalent form as

$$\bar{\theta}_4(\psi,q) = \sum_{k=0}^\infty \sum_{j=0}^\infty \frac{(-1)^j(-\psi)^k(P_{eff})^{\frac{k}{2}}(\alpha)^j\Gamma(\frac{k}{2}+j)}{k!j!(1-\alpha)^{\frac{k}{2}+j}\Gamma(\frac{k}{2})}\cdot\frac{1}{q^j}. \tag{22}$$

Employing inverse Laplace transformation, we get

$$\theta_4(\psi,\eta) = \sum_{k=0}^\infty \sum_{j=0}^\infty \frac{(-1)^j(-\psi)^k(P_{eff})^{\frac{k}{2}}(\alpha)^j\Gamma(\frac{k}{2}+j)}{k!j!(1-\alpha)^{\frac{k}{2}+j}\Gamma(\frac{k}{2})}\cdot\frac{\eta^{j-1}}{\Gamma(j)}. \tag{23}$$

### 4.2. Exact Solution of Heat Profile with ABC Time Fractional Derivative

The fractional model for Equation (9) formulated on the base of the ABC fractional time derivative is provided as:

$$\frac{d^2\theta(\psi,\eta)}{d\psi^2} - Pr_{eff} {}^{ABC}D_\eta^\alpha \theta(\psi,\eta) = 0. \tag{24}$$

Applying the Laplace transformation technique to write the solution of Equation (24) with conditions as Equations (11)–(13); we have

$$\frac{d^2\bar{\theta}(\psi,q)}{d\psi^2} - Pr_{eff}\frac{q^\alpha}{(1-\alpha)q^\alpha + \alpha}\bar{\theta}(\psi,q) = 0. \tag{25}$$

with

$$\bar{\theta}(0,q) = \frac{1}{q} - \frac{a}{q+b} \quad \text{and} \quad \bar{\theta}(\psi,q) \to 0 \ \text{as} \ \psi \to \infty. \tag{26}$$

and its general solution has the form

$$\bar{\theta}(\psi,q) = \chi_3 e^{\psi\sqrt{\frac{q^\alpha Pr_{eff}}{(1-\alpha)q^\alpha+\alpha}}} + \chi_4 e^{-\psi\sqrt{\frac{q^\alpha Pr_{eff}}{(1-\alpha)q^\alpha+\alpha}}}. \tag{27}$$

we applied the conditions for temperature mentioned in Equation (26) to determine the unknown constants $\chi_3$ and $\chi_4$; we get

$$\bar{\theta}(\psi,q) = \left(\frac{1}{q} - \frac{a}{q+b}\right)e^{-\psi\sqrt{\frac{q^\alpha Pr_{eff}}{(1-\alpha)q^\alpha+\alpha}}}. \tag{28}$$

Moreover, we can write

$$\bar{\theta}(\psi,q) = \bar{\theta}_5(\psi,q) - a\bar{\theta}_6(\psi,q). \tag{29}$$

to get the required solution of Equation (29), using Laplace inverse transformation, which is written as:

$$\theta(\psi,\eta) = \theta_5(\psi,\eta) - a\theta_6(\psi,\eta). \tag{30}$$

where

$$\bar{\theta}_5(\psi,q) = \frac{1}{q}e^{-\psi\sqrt{\frac{q^\alpha Pr_{eff}}{(1-\alpha)q^\alpha+\alpha}}}. \tag{31}$$

It is difficult to find $\theta_5(\psi,\eta)$ concerning the exponential form, so we express $\bar{\theta}_5(\psi,q)$ in its equivalent form as

$$\bar{\theta}_5(\psi,q) = \sum_{k=0}^{\infty}\sum_{j=0}^{\infty}\frac{(-1)^j(-\psi)^k(Pr_{eff})^{\frac{k}{2}}(\alpha)^j\Gamma(\frac{k}{2}+j)}{k!j!(1-\alpha)^{\frac{k}{2}+j}\Gamma(\frac{k}{2})}\cdot\frac{1}{q^{j\alpha+1}}. \tag{32}$$

Employing the inverse Laplace transformation, we get

$$\theta_5(\psi,\eta) = \sum_{k=0}^{\infty}\sum_{j=0}^{\infty}\frac{(-1)^j(-\psi)^k(Pr_{eff})^{\frac{k}{2}}(\alpha)^j\Gamma(\frac{k}{2}+j)}{k!j!(1-\alpha)^{\frac{k}{2}+j}\Gamma(\frac{k}{2})}\cdot\frac{\eta^{j\alpha}}{\Gamma(j\alpha+1)}. \tag{33}$$

$$\theta_6(\psi, \eta) = (\theta_7 * \theta_8)(\eta),$$

$$\theta_7(\psi, \eta) = \mathcal{L}^{-1}\left\{\frac{1}{q+b}\right\} = e^{-b\eta},$$

$$\bar{\theta}_8(\psi, q) = e^{-\psi\sqrt{\frac{q^\alpha P_{reff}}{(1-\alpha)q^\alpha + \alpha}}}, \tag{34}$$

Similarly, concerning Equation (31), to compute $\theta_8(\psi, \eta)$ we express $\bar{\theta}_8(\psi, q)$ in its series equivalent form as

$$\bar{\theta}_8(\psi, q) = \sum_{k=0}^{\infty}\sum_{j=0}^{\infty}\frac{(-1)^j(-\psi)^k(P_{reff})^{\frac{k}{2}}(\alpha)^j\Gamma(\frac{k}{2}+j)}{k!j!(1-\alpha)^{\frac{k}{2}+j}\Gamma(\frac{k}{2})}\cdot\frac{1}{q^{j\alpha}}. \tag{35}$$

Employing inverse Laplace transformation, we get

$$\theta_8(\psi, \eta) = \sum_{k=0}^{\infty}\sum_{j=0}^{\infty}\frac{(-1)^j(-\psi)^k(P_{reff})^{\frac{k}{2}}(\alpha)^j\Gamma(\frac{k}{2}+j)}{k!j!(1-\alpha)^{\frac{k}{2}+j}\Gamma(\frac{k}{2})}\cdot\frac{\eta^{j\alpha-1}}{\Gamma(j\alpha)}. \tag{36}$$

*4.3. Exact Solution of Mass Profile with CF Time Fractional Derivative*

The fractional model for Equation (10) formulated on the base of Caputo–Fabrizio fractional time derivative is provided as:

$$\frac{d^2C(\psi, \eta)}{d\psi^2} - S_c\left({}^{CF}D_\eta^\alpha + R_c\right)C(\psi, \eta) = 0. \tag{37}$$

Solving Equation (37) using (11)–(13) and employing Laplace Transformation technique, the resulting equations are written as:

$$\frac{d^2\bar{C}(\psi, q)}{d\psi^2} - S_c\left(\frac{q}{(1-\alpha)q + \alpha} + R_c\right)\bar{C}(\psi, q) = 0. \tag{38}$$

with

$$\bar{C}(0, q) = \frac{1}{q} \quad \text{and} \quad \bar{C}(\psi, q) \to 0 \text{ as } \psi \to \infty. \tag{39}$$

The solution in general form is

$$\bar{C}(\psi, q) = \chi_5 e^{\psi\sqrt{S_c\left(\frac{q}{(1-\alpha)q+\alpha}+R_c\right)}} + \chi_6 e^{-\psi\sqrt{S_c\left(\frac{q}{(1-\alpha)q+\alpha}+R_c\right)}}. \tag{40}$$

Concerning the values of constants $\chi_5$ and $\chi_6$, conditions for concentration are implemented given Equation (39), so

$$\bar{C}(\psi, q) = \frac{1}{q}e^{-\psi\sqrt{S_c\left(\frac{q}{(1-\alpha)q+\alpha}+R_c\right)}}. \tag{41}$$

It is complicated to find $C(\psi, \eta)$ from the exponential form, so $\bar{C}(\psi, q)$ in its equivalent form is:

$$\bar{C}(\psi, q) = \sum_{k=0}^{\infty}\sum_{m=0}^{\infty}\sum_{n=0}^{\infty}\frac{(-\psi)^k(-1)^n(R_c)^{\frac{k}{2}-m}(S_c)^{\frac{k}{2}}\alpha^n\Gamma(\frac{k}{2}+1)\Gamma(m+n)}{k!m!n!(1-\alpha)^{m+n}\Gamma(\frac{k}{2}-m+1)\Gamma(m)}\cdot\frac{1}{q^{1+n}}. \tag{42}$$

The required solution for the diffusion equation, after taking the Laplace inverse, is written as:

$$C(\psi,\eta) = \sum_{k=0}^{\infty} \sum_{m=0}^{\infty} \sum_{n=0}^{\infty} \frac{(-\psi)^k(-1)^n(R_c)^{\frac{k}{2}-m}(S_c)^{\frac{k}{2}}\alpha^n\Gamma(\frac{k}{2}+1)\Gamma(m+n)}{k!m!n!(1-\alpha)^{m+n}\Gamma(\frac{k}{2}-m+1)\Gamma(m)} \cdot \frac{\eta^n}{\Gamma(1+n)}. \tag{43}$$

### 4.4. Exact Solution of Mass Profile with ABC Time Fractional Derivative

The fractional model for Equation (10) formulated on the base of the ABC fractional time derivative is provided as:

$$\frac{d^2 C(\psi,\eta)}{d\psi^2} - S_c\left(^{ABC}D_\eta^\alpha + R_c\right)C(\psi,\eta) = 0. \tag{44}$$

Solving Equation (44) using (11)–(13) and employing the Laplace transformation technique, the resulting equations are written as:

$$\frac{d^2 \bar{C}(\psi,q)}{d\psi^2} - S_c\left(\frac{q^\alpha}{(1-\alpha)q^\alpha+\alpha} + R_c\right)\bar{C}(\psi,q) = 0. \tag{45}$$

with

$$\bar{C}(0,q) = \frac{1}{q} \quad \text{and} \quad \bar{C}(\psi,q) \to 0 \ \text{as} \ \psi \to \infty. \tag{46}$$

The solution in general form is

$$\bar{C}(\psi,q) = \chi_7 e^{\psi\sqrt{S_c\left(\frac{q^\alpha}{(1-\alpha)q^\alpha+\alpha}+R_c\right)}} + \chi_8 e^{-\psi\sqrt{S_c\left(\frac{q^\alpha}{(1-\alpha)q^\alpha+\alpha}+R_c\right)}}. \tag{47}$$

Concerning the values of constants $\chi_7$ and $\chi_8$, conditions for concentration are implemented given Equation (39), so

$$\bar{C}(\psi,q) = \frac{1}{q}e^{-\psi\sqrt{S_c\left(\frac{q^\alpha}{(1-\alpha)q^\alpha+\alpha}+R_c\right)}}. \tag{48}$$

It is complicated to find $C(\psi,\eta)$ from the exponential form, so $\bar{C}(\psi,q)$ in its equivalent form is written as

$$\bar{C}(\psi,q) = \sum_{k=0}^{\infty} \sum_{m=0}^{\infty} \sum_{n=0}^{\infty} \frac{(-\psi)^k(-1)^n(R_c)^{\frac{k}{2}-m}(S_c)^{\frac{k}{2}}\alpha^n\Gamma(\frac{k}{2}+1)\Gamma(m+n)}{k!m!n!(1-\alpha)^{m+n}\Gamma(\frac{k}{2}-m+1)\Gamma(m)} \cdot \frac{1}{q^{\alpha n+1}}. \tag{49}$$

The required solution for the diffusion equation, after taking the Laplace inverse, is written as:

$$C(\psi,\eta) = \sum_{k=0}^{\infty} \sum_{m=0}^{\infty} \sum_{n=0}^{\infty} \frac{(-\psi)^k(-1)^n(R_c)^{\frac{k}{2}-m}(S_c)^{\frac{k}{2}}\alpha^n\Gamma(\frac{k}{2}+1)\Gamma(m+n)}{k!m!n!(1-\alpha)^{m+n}\Gamma(\frac{k}{2}-m+1)\Gamma(m)} \cdot \frac{\eta^{\alpha n}}{\Gamma(\alpha n+1)}. \tag{50}$$

### 4.5. Exact Solution of Velocity Profile with CF Time Fractional Derivative

The fractional model for Equation (8) formulated on the base of the Caputo–Fabrizio fractional time derivative is provided as:

$$^{CF}D_\eta^\alpha \zeta(\psi,q) = \frac{d^2 \zeta(\psi,\eta)}{d\psi^2} + \theta(\psi,\eta)\cos(v) + NC(\psi,\eta)\cos(v) - K\zeta(\psi,\eta) - M\sin^2(\hbar)\zeta(\psi,\eta). \tag{51}$$

Implementing the technique of the Laplace transformation and making use of the concerned stated conditions concerning Equation (51), the differential equation is obtained as:

$$\left(\frac{q}{(1-\alpha)q+\alpha}\right)\bar{\zeta}(\psi,q) = \frac{d^2 \bar{\zeta}(\psi,q)}{d\psi^2} + \bar{\theta}(\psi,q)\cos(v) + N\bar{C}(\psi,q)\cos(v) - K\bar{\zeta}(\psi,q) - M\sin^2(\hbar)\bar{\zeta}(\psi,q), \tag{52}$$

with the stated conditions for the considered problem

$$\bar{\zeta}(0,q) = F(q) \quad \text{and} \quad \bar{\zeta}(\psi,q) \to 0 \quad \text{as} \quad \psi \to \infty, \tag{53}$$

where $\bar{\zeta}(\psi,q)$ and $F(q)$ denote the Laplace transformations for $\zeta(\psi,\eta)$ and $f(\eta)$, respectively. Substituting $\bar{\theta}(\psi,q)$ and $\overline{C}(\psi,q)$, as mentioned in Equation (18) and Equation (41) into Equation (52), it can be expressed in more precise form

$$\frac{d^2\bar{\zeta}(\psi,q)}{d\psi^2} - \left(\frac{b_2 q + c_2}{q + c_1}\right)\bar{\zeta}(\psi,q) = -\cos(v)\left(\frac{1}{q} - \frac{a}{q+b}\right)e^{-\psi\sqrt{\frac{b_1 q \mathrm{Pr}_{eff}}{q+c_1}}} - \frac{N\cos(v)}{q}e^{-\psi\sqrt{\frac{b_1 q S_c}{q+c_1}+R_c}}. \tag{54}$$

The obtained solution of Equation (54) with the conditions mentioned in Equation (53) is represented as:

$$\bar{\zeta}(\psi,q) = F(q)e^{-\psi\sqrt{\frac{b_2 q + c_2}{q+c_1}}} + \cos(v)\left(\frac{q+c_1}{b_3 q + c_2}\right)\left(\frac{1}{q} - \frac{a}{q+b}\right)\left[e^{-\psi\sqrt{\frac{b_1 q \mathrm{Pr}_{eff}}{q+c_1}}} - e^{-\psi\sqrt{\frac{b_2 q + c_2}{q+c_1}}}\right]$$

$$+ N\cos(v)\left(\frac{q+c_1}{b_4 q + c_3}\right)\frac{1}{q}\left[e^{-\psi\sqrt{\frac{b_1 q S_c}{q+c_1}+R_c}} - e^{-\psi\sqrt{\frac{b_2 q + c_2}{q+c_1}}}\right]. \tag{55}$$

Now, taking Laplace inverse, the solution of Equation (55), for velocity field, is written as:

$$\zeta(\psi,\eta) = \int_0^\eta \omega_1(\psi,s)\cdot f(\eta-s)ds + \frac{1}{b_3}\cos(v)\left[\theta(\psi,\eta) - \omega_2(\psi,\eta) + a\int_0^\eta e^{-bs}\omega_1(\psi,\eta-s)ds\right]$$

$$+ \frac{c_7}{b_3}\cos(v)\left[\int_0^\eta e^{-c_6 s}\theta(\psi,\eta-s)ds - \int_0^\eta e^{-c_6 s}\omega_2(\psi,\eta-s)ds + a\int_0^\eta\int_0^s e^{-c_6(\eta-s)}\omega_1(\psi,s-u)e^{-bu}duds\right]$$

$$+ \frac{N}{b_4}\cos(v)[C(\psi,\eta) - \omega_2(\psi,\eta)] + \frac{Nc_9}{b_4}\cos(v)\left[\int_0^\eta e^{-c_8 s}C(\psi,\eta-s)ds - \int_0^\eta e^{-c_8 s}\omega_2(\psi,\eta-s)ds\right], \tag{56}$$

### 4.6. Exact Solution of Velocity Profile with ABC Time Fractional Derivative

The fractional model for Equation (8) formulated on the base of ABC fractional time derivative is provided as:

$$^{ABC}D_\eta^\alpha\zeta(\psi,q) = \frac{d^2\zeta(\psi,\eta)}{d\psi^2} + \theta(\psi,\eta)\cos(v) + NC(\psi,\eta)\cos(v) - K\zeta(\psi,\eta) - M\sin^2(\hbar)\zeta(\psi,\eta). \tag{57}$$

Implementing the technique of Laplace transformation and making use of the concerned stated conditions for Equation (57), the differential equation is obtained as:

$$\left(\frac{q^\alpha}{(1-\alpha)q^\alpha + \alpha}\right)\bar{\zeta}(\psi,q) = \frac{d^2\bar{\zeta}(\psi,q)}{d\psi^2} + \bar{\theta}(\psi,q)\cos(v) + N\overline{C}(\psi,q)\cos(v) - K\bar{\zeta}(\psi,q) - M\sin^2(\hbar)\bar{\zeta}(\psi,q), \tag{58}$$

with the stated conditions for the considered problem

$$\bar{\zeta}(0,q) = F(q) \quad \text{and} \quad \bar{\zeta}(\psi,q) \to 0 \quad \text{as} \quad \psi \to \infty, \tag{59}$$

where $\bar{\zeta}(\psi,q)$ and $F(q)$ denote the Laplace transformations for $\zeta(\psi,\eta)$ and $f(\eta)$, respectively. Substituting $\bar{\theta}(\psi,q)$ and $\overline{C}(\psi,q)$, as mentioned in Equation (28) and Equation (48) into Equation (58), it can be expressed in more precise form

$$\frac{d^2\bar{\zeta}(\psi,q)}{d\psi^2} - \left(\frac{b_2 q^\alpha + c_2}{q^\alpha + c_1}\right)\bar{\zeta}(\psi,q) = -\cos(v)\left(\frac{1}{q} - \frac{a}{q+b}\right)e^{-\psi\sqrt{\frac{b_1 q^\alpha \mathrm{Pr}_{eff}}{q^\alpha+c_1}}} - \frac{N\cos(v)}{q}e^{-\psi\sqrt{\frac{b_1 q^\alpha S_c}{q^\alpha+c_1}+R_c}}. \tag{60}$$

The obtained solution of Equation (60) with the conditions mentioned in Equation (59) is represented as:

$$
\bar{\zeta}(\psi,q) = F(q)e^{-\psi\sqrt{\frac{b_2 q^\alpha + c_2}{q^\alpha + c_1}}} + \cos(v)\left(\frac{q^\alpha + c_1}{b_3 q^\alpha + c_2}\right)\left(\frac{1}{q} - \frac{a}{q+b}\right)\left[e^{-\psi\sqrt{\frac{b_1 q^\alpha \mathrm{Pr}_{eff}}{q^\alpha + c_1}}} - e^{-\psi\sqrt{\frac{b_2 q^\alpha + c_2}{q^\alpha + c_1}}}\right]
$$

$$
+ N\cos(v)\left(\frac{q^\alpha + c_1}{b_4 q^\alpha + c_3}\right)\frac{1}{q}\left[e^{-\psi\sqrt{\frac{b_1 q^\alpha S_c + R_c}{q^\alpha + c_1}}} - e^{-\psi\sqrt{\frac{b_2 q^\alpha + c_2}{q^\alpha + c_1}}}\right]. \tag{61}
$$

Now, taking Laplace inverse, the solution of Equation (61), for velocity field, is written as:

$$
\zeta(\psi,\eta) = \int_0^\eta \omega_3(\psi,s)\cdot f(\eta - s)ds + \frac{1}{b_3}\cos(v)\left[\theta(\psi,\eta) - \omega_4(\psi,\eta) + a\int_0^\eta e^{-bs}\omega_3(\psi,\eta - s)ds\right] + \frac{c_7}{b_3}\cos(v)
$$

$$
\times \left[\int_0^\eta F_\alpha(-c_6 s)\theta(\psi,\eta - s)ds - \int_0^\eta F_\alpha(-c_6 s)\omega_2(\psi,\eta - s)ds + a\int_0^\eta\int_0^s e^{-b(\eta - s)}\omega_1(\psi,s - u)F_\alpha(-c_6 u)duds\right]
$$

$$
+ \frac{N}{b_4}\cos(v)[C(\psi,\eta) - \omega_4(\psi,\eta)] + \frac{Nc_9}{b_4}\cos(v)\left[\int_0^\eta F_\alpha(-c_8 s)C(\psi,\eta - s)ds - \int_0^\eta F_\alpha(-c_8 s)\omega_2(\psi,\eta - s)ds\right], \tag{62}
$$

Furthermore, in the above expressions

$$
\omega_1(\psi,\eta) = \mathcal{L}^{-1}\left\{e^{-\psi\sqrt{\frac{b_2 q + c_2}{q + c_1}}}\right\}
$$

$$
= \sum_{n=0}^\infty\sum_{m=0}^\infty\sum_{p=0}^\infty \frac{(-1)^p(-\psi)^n\cdot(b_2)^{\frac{n}{2}}\cdot(c_5)^m\cdot(c_1)^p\cdot\Gamma(\frac{n}{2}+1)\Gamma(m+p)}{n!m!p!(m-1)!\cdot\Gamma(\frac{n}{2}-m+1)}\cdot\frac{(\eta)^{m+p-1}}{\Gamma(m+p)},
$$

$$
\omega_2(\psi,\eta) = \mathcal{L}^{-1}\left\{\frac{e^{-\psi\sqrt{\frac{b_2 q + c_2}{q + c_1}}}}{q}\right\}
$$

$$
= \sum_{n=0}^\infty\sum_{m=0}^\infty\sum_{p=0}^\infty \frac{(-1)^p(-\psi)^n\cdot(b_2)^{\frac{n}{2}}\cdot(c_5)^m\cdot(c_1)^p\cdot\Gamma(\frac{n}{2}+1)\Gamma(m+p)}{n!m!p!(m-1)!\cdot\Gamma(\frac{n}{2}-m+1)}\cdot\frac{(\eta)^{m+p}}{\Gamma(m+p+1)},
$$

$$
\omega_3(\psi,\eta) = \mathcal{L}^{-1}\left\{e^{-\psi\sqrt{\frac{b_2 q^\alpha + c_2}{q^\alpha + c_1}}}\right\}
$$

$$
= \sum_{n=0}^\infty\sum_{k=0}^\infty \frac{(-\psi)^n\cdot(b_2)^{\frac{n}{2}}\cdot(c_4)^{\frac{n}{2}-k}\cdot\Gamma(\frac{n}{2}+1)}{n!k!\Gamma(\frac{n}{2}-k+1)}\cdot G_{\alpha,\alpha k,\frac{n}{2}}(-c_1,t),
$$

$$
\omega_4(\psi,\eta) = \mathcal{L}^{-1}\left\{\frac{e^{-\psi\sqrt{\frac{b_2 q^\alpha + c_2}{q^\alpha + c_1}}}}{q}\right\}
$$

$$
= \sum_{n=0}^\infty\sum_{k=0}^\infty \frac{(-\psi)^n\cdot(b_2)^{\frac{n}{2}}\cdot(c_4)^{\frac{n}{2}-k}\cdot\Gamma(\frac{n}{2}+1)}{n!k!\Gamma(\frac{n}{2}-k+1)}\cdot G_{\alpha,\alpha k-1,\frac{n}{2}}(-c_1,t),
$$

Moreover,

$$b_1 = \frac{1}{1-\alpha}, \quad b_2 = b_1 + K + M\sin^2(\hbar), \quad b_3 = b_2 - b_1 Pr_e ff, \quad b_4 = b_2 - b_1 S_c - R_c,$$

$$c_1 = \frac{\alpha}{1-\alpha}, \quad c_2 = c_1(K + M\sin^2(\hbar)), \quad c_3 = c_2 - c_1 R_c, \quad c_4 = \frac{c_2}{b_2}, \quad c_5 = c_4 - c_1,$$

$$c_6 = \frac{c_2}{b_3}, \quad c_7 = c_1 - c_6, \quad c_8 = \frac{c_3}{b_4}, \quad c_9 = c_1 - c_8, \quad \text{and} \quad (\wp * \chi)(t) = \int_0^t \wp(t)\chi(t-\tau)d\tau.$$

The function $G_{h,b,l}(.,\tau)$ used in the above expressions is known as the Generalized Lorenzo Hartly function. The function $F_{-a\eta}$ is known as the Robotnov and Hartley function. The Laplace inverse of these functions is defined as:

$$\mathcal{L}^{-1}\left\{\frac{s^b}{(s^h - j)^l}\right\} = G_{h,b,l}(j,\tau);$$

$$Re(hl - b) > 0,$$

$$Re(s) > 0,$$

$$\left|\frac{j}{s^h}\right| < 1$$

and

$$\mathcal{L}^{-1}\left\{\frac{1}{q^\beta + \alpha}\right\} = F_\beta(-\alpha\eta); \beta > 0$$

Now, in the next section, we shall explore the fluid dynamics under the effect of oscillating motion or slow acceleration of the plate coupled with the objective for deep understanding of the physical aspects of the acquired results.

## 5. Various Cases Concerning the Motion of the Plate

We will establish the solution expression relative to motions generated due to oscillation of the plate and slow acceleration in the plate (when $\gamma < 1$).

### 5.1. Case-I: $f(\eta) = H(\eta)\eta^\gamma$ (for Variable Accelerating Plate)

Now, substituting $f(\eta) = H(\eta)\eta^\gamma$, with $\gamma > 0$, into Equation (56), we get

$$\zeta(\psi,\eta) = \int_0^\eta \omega_1(\psi,s)\cdot(\eta - s)^\gamma ds + \frac{1}{b_3}\cos(v)\left[\theta(\psi,\eta) - \omega_2(\psi,\eta) + a\int_0^\eta e^{-bs}\omega_1(\psi,\eta - s)ds\right]$$

$$+ \frac{c_7}{b_3}\cos(v)\left[\int_0^\eta e^{-c_6 s}\theta(\psi,\eta - s)ds - \int_0^\eta e^{-c_6 s}\omega_2(\psi,\eta - s)ds + a\int_0^\eta\int_0^s e^{-c_6(\eta - s)}\omega_1(\psi,s - u)e^{-bu}duds\right]$$

$$+ \frac{N}{b_4}\cos(v)[C(\psi,\eta) - \omega_2(\psi,\eta)] + \frac{Nc_9}{b_4}\cos(v)\left[\int_0^\eta e^{-c_8 s}C(\psi,\eta - s)ds - \int_0^\eta e^{-c_8 s}\omega_2(\psi,\eta - s)ds\right], \quad (63)$$

which represents motion of fluid caused by highly, slowly or constantly accelerating plate. Additionally, consider the case for $\gamma = 0$, i.e., when $f(\eta) = H(\eta)$ is

$$
\zeta(\psi, \eta) = \int_0^\eta \omega_1(\psi, s)ds + \frac{1}{b_3} \cos(v) \left[ \theta(\psi, \eta) - \omega_2(\psi, \eta) + a \int_0^\eta e^{-bs} \omega_1(\psi, \eta - s)ds \right]
$$

$$
+ \frac{c_7}{b_3} \cos(v) \left[ \int_0^\eta e^{-c_6 s} \theta(\psi, \eta - s)ds - \int_0^\eta e^{-c_6 s} \omega_2(\psi, \eta - s)ds + a \int_0^\eta \int_0^s e^{-c_6(\eta - s)} \omega_1(\psi, s - u)e^{-bu}du\,ds \right]
$$

$$
+ \frac{N}{b_4} \cos(v)[C(\psi, \eta) - \omega_2(\psi, \eta)] + \frac{Nc_9}{b_4} \cos(v) \left[ \int_0^\eta e^{-c_8 s} C(\psi, \eta - s)ds - \int_0^\eta e^{-c_8 s} \omega_2(\psi, \eta - s)ds \right], \tag{64}
$$

*5.2. Case-II: $f(\eta) = \cos(\omega\eta)\,H(\eta)$ or $\sin(\omega\eta)H(\eta)$ (for Oscillating Plate)*

Putting $f(\eta) = \cos(\omega\eta)\,H(\eta)$ or $\sin(\omega\eta)H(\eta)$ into Equation (56), we obtain

$$
\zeta_c(\psi, \eta) = \int_0^\eta \omega_1(\psi, s) \cos(\omega(\eta - s))ds + \frac{1}{b_3} \cos(v) \left[ \theta(\psi, \eta) - \omega_2(\psi, \eta) + a \int_0^\eta e^{-bs} \omega_1(\psi, \eta - s)ds \right]
$$

$$
+ \frac{c_7}{b_3} \cos(v) \left[ \int_0^\eta e^{-c_6 s} \theta(\psi, \eta - s)ds - \int_0^\eta e^{-c_6 s} \omega_2(\psi, \eta - s)ds + a \int_0^\eta \int_0^s e^{-c_6(\eta - s)} \omega_1(\psi, s - u)e^{-bu}du\,ds \right]
$$

$$
+ \frac{N}{b_4} \cos(v)[C(\psi, \eta) - \omega_2(\psi, \eta)] + \frac{Nc_9}{b_4} \cos(v) \left[ \int_0^\eta e^{-c_8 s} C(\psi, \eta - s)ds - \int_0^\eta e^{-c_8 s} \omega_2(\psi, \eta - s)ds \right], \tag{65}
$$

$$
\zeta_s(\psi, \eta) = \int_0^\eta \omega_1(\psi, s) \sin(\omega(\eta - s))ds + \frac{1}{b_3} \cos(v) \left[ \theta(\psi, \eta) - \omega_2(\psi, \eta) + a \int_0^\eta e^{-bs} \omega_1(\psi, \eta - s)ds \right]
$$

$$
+ \frac{c_7}{b_3} \cos(v) \left[ \int_0^\eta e^{-c_6 s} \theta(\psi, \eta - s)ds - \int_0^\eta e^{-c_6 s} \omega_2(\psi, \eta - s)ds + a \int_0^\eta \int_0^s e^{-c_6(\eta - s)} \omega_1(\psi, s - u)e^{-bu}du\,ds \right]
$$

$$
+ \frac{N}{b_4} \cos(v)[C(\psi, \eta) - \omega_2(\psi, \eta)] + \frac{Nc_9}{b_4} \cos(v) \left[ \int_0^\eta e^{-c_8 s} C(\psi, \eta - s)ds - \int_0^\eta e^{-c_8 s} \omega_2(\psi, \eta - s)ds \right], \tag{66}
$$

## 6. Results Validation

In order to discuss the validation of our derived results, we take $\hbar = \frac{\pi}{2}$, $K = 0$, $v = 0$ and $\alpha \to 1$ in Equations (56) and (62) then recover the corresponding equations as Shah et al. [15] obtained for the viscous fluid case. Furthermore, when $f(\eta) = H(\eta)$ (the Heaviside unit step function) $K = 0, v = 0$ and $\hbar = \frac{\pi}{2}$ with $\alpha \to 1$ in relations (56) and (62). The achieved solution expressions are the same as those derived by Narahari and Debnath [13] (Equation (11-a) taking $a = 0$) and also Tokis [38] (Equation (12)) for the case when thermal and concentration effects are ignored. Evidently, by adjusting $f(\cdot)$ in different appropriate forms, the exact solution of fluid motion of theses types are recovered. This proves the authenticity of our newly established solution expressions.

## 7. Results and Discussion

In this work, the general equations of double diffusive magneto-free convection for viscous fluid are presented in non-dimensional form, and are applied to a moving heated vertical plate as in the boundary layer flow up, with the existence of an externally magnetic field which is either moving or fixed consistent with the plate. The thermal transport phenomenon is discussed in the presence of constant concentration coupled with the first order chemical reaction with exponential heating. An innovative definition of CF and

ABC time fractional operators is implemented to hypothesize the constitutive mass, energy and momentum equations. The solutions based on special functions are obtained using the Laplace transform method to tackle the non-dimensional partial differential equations for velocity, mass and energy. Moreover, the heat transfer aspects, flow dynamics and their credence on the parameters, involved in the problem like effective Prandtl number '$Pr_{eff}$', dimensionless time '$\eta$', Schmidt number '$Sc$', Ratio of buoyancy forces $N$, fractional parameter $\alpha$ and chemical reaction parameter $R_c$, are drawn out by graphical illustrations. The analytically derived results are summarized in Figures 1–11; the behavior of fluid velocity is discussed for four different values of $\alpha = 0.1, 0.3, 0.6, 0.9$ along with involving physical parameters, by considering $a = 0.70$, $b = 0.10$, $\eta = 0.8$, $N = 0.25$, $\hbar = \frac{\pi}{2}$, $M = 0.6$, $Pr_{eff} = 0.71$, $Sc = 0.6$, $K = 0.3$, $R_c = 0.7$, $\gamma = 0.7$ for the function $f(\eta) = H(\eta)\eta^{\gamma}$.

Figures 1 and 2 represent the influence of $Pr_{eff}$ via CF and ABC with $\alpha = 0.1, 0.3, 0.6, 0.9$, on fluid temperature. It is indicated that advances in Prandtl number reduce the temperature profile of the moving fluid for distinct values of fractional parameter. The boundary layer of temperature profile gets thicker due to the fact that there is a small rate of thermal diffusion and the resultant temperature profile decreases.

Concerning Figures 3 and 4, the action of $Sc$ on concentration profile along with the applications of CF and ABC fractional operators are analyzed for various values of fractional parameter as $\alpha = 0.1, 0.3, 0.6, 0.9$. It is remarked that the concentration profile of the fluid will decrease as $Sc$ increases. Physically, the reduction in the boundary layer of concentration happened to correspond to enhancing the Schmidt number.

Concerning Figures 5 and 6, the action of $Sc$ on the velocity profile along with the applications of CF and ABC fractional operators are analyzed for various values of fractional parameter as $\alpha = 0.1, 0.3, 0.6, 0.9$. It is remarked that the velocity of the fluid flow will decrease as $Sc$ increases. Physically, the Schmidt number $Sc$ is mathematically defined as the ratio of momentum to mass diffusivity. It is a fact that the layer of momentum diffusivity of the fluid is more viscous; as a result velocity decreases. Schmidt number $Sc$ signified concentration and velocity in free convection flow regimes of fluids concern relative effectiveness momentum.

Figures 7 and 8 represent the effects of $N$ by the application of CF and ABC fractional operators on the fluid velocity at time $\eta = 1.4$ for both aiding and opposing flows by considering various values of $\alpha$. For adding flows $N > 0$, the thermal buoyancy force is supported by the buoyancy force (caused by species diffusion), which results in the rise in the velocity for the rise in the values of $N$. For opposing flows $N < 0$, the buoyancy force results in a reversal flow effect and hence resists the flow of fluid.

Figures 9 and 10 portray the behavior of $R_c$ on fluid velocity via CF and ABC time fractional operators for distinct values of $\alpha$ and take time as fixed for both operators. It is realized that the velocity decreases corresponding to the increase in the value of $R_c$. Further, it is pointed out that the momentum profiles describe the same behaviour for both the fractional derivative operators.

Figures 11 and 12 illustrate the variation in velocity field via CF and ABC time fractional operators for different values of $\alpha$ with $\eta = 0.8, 1.6, 2.6$; it can be claimed that the velocity profile is elevated corresponding to large values of time. Moreover, we see that the smooth decline in the velocity from definite large values at the endpoint to the asymptotical value is enlarged.

Figure 13 illustrates the impacts of time $\eta$ with $\alpha = 0.20, 0.40, 0.60, 0.80$ on velocity field with the application of CF and ABC time fractional operators, it can be observed that the velocity profile is elevated corresponding to large values of time. Moreover, a comparison is conducted between the contours portrayed for velocity profile via CF and ABC. It is remarked that the velocity curve with non-integer operator ABC is superior as compared to CF time fractional operator.

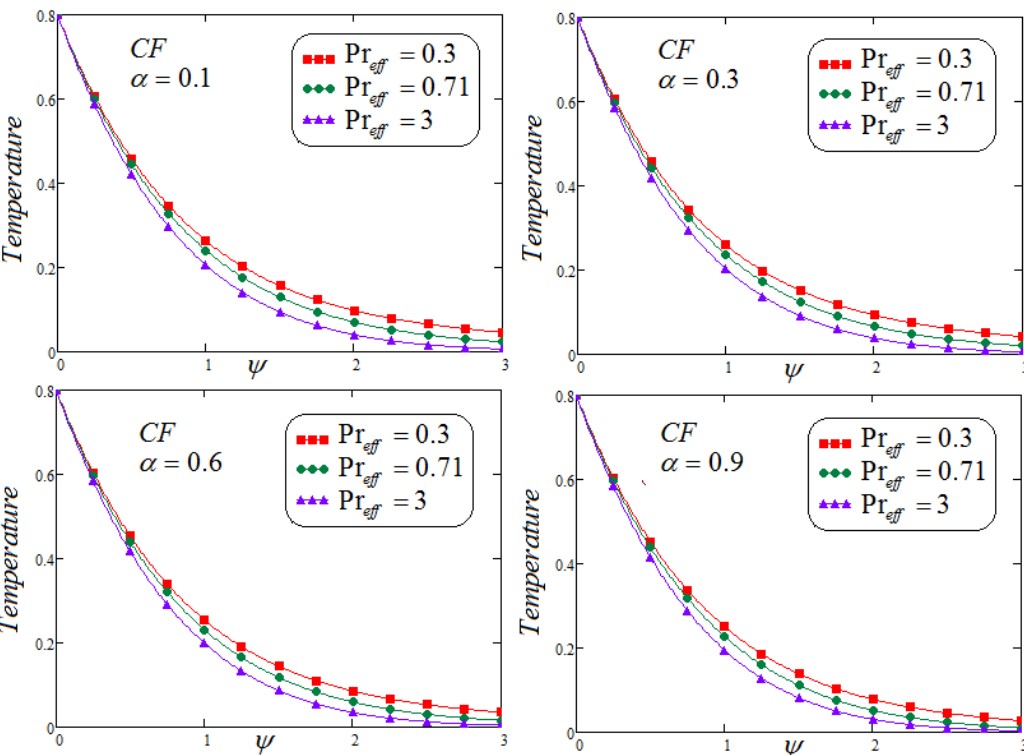

**Figure 1.** Trace of dimensionless temperature for dissimilar values of $Pr_{eff}$ via CF.

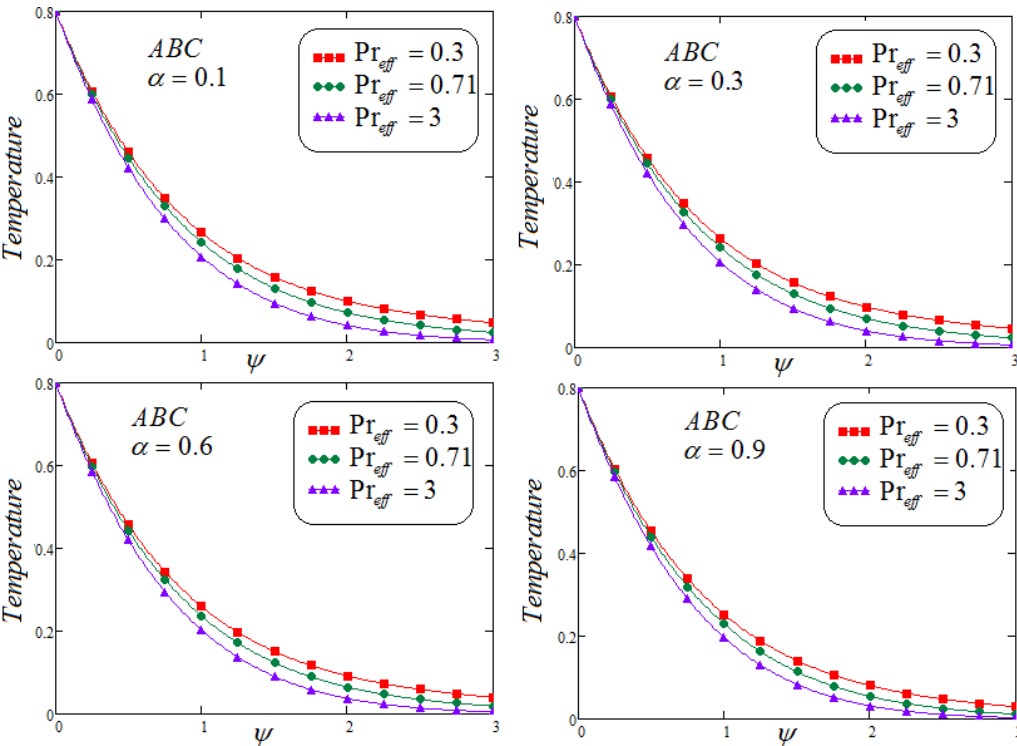

**Figure 2.** Trace of dimensionless temperature for dissimilar values of $Pr_{eff}$ via ABC.

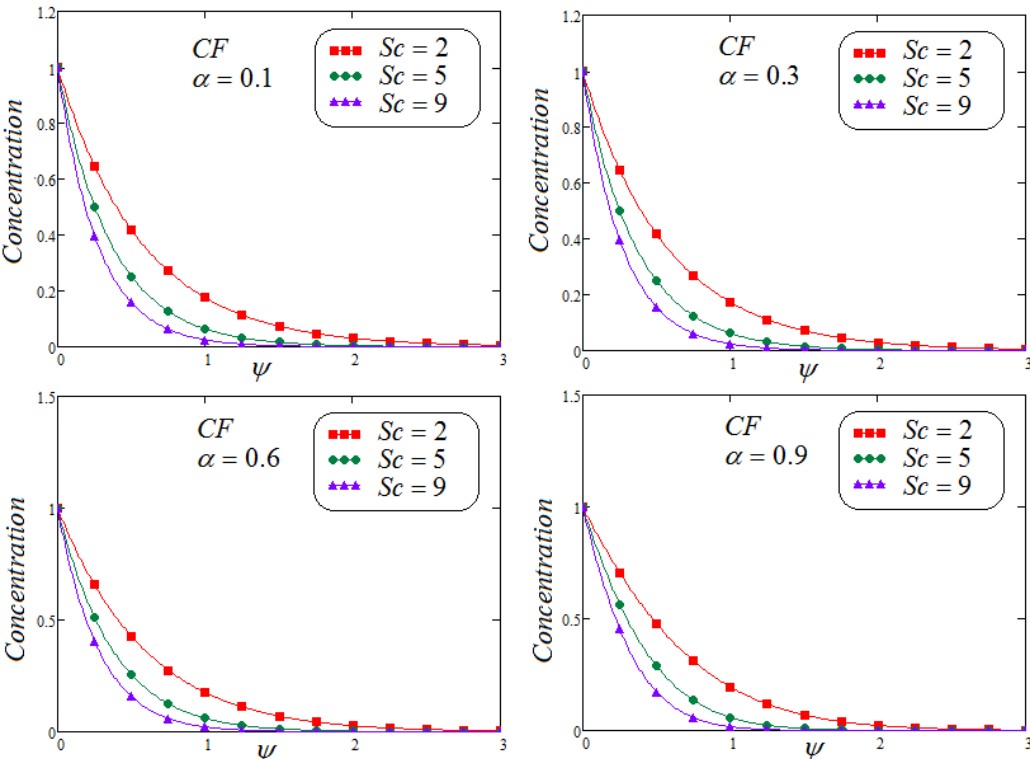

**Figure 3.** Trace of dimensionless concentration for dissimilar values of $Sc$ via CF.

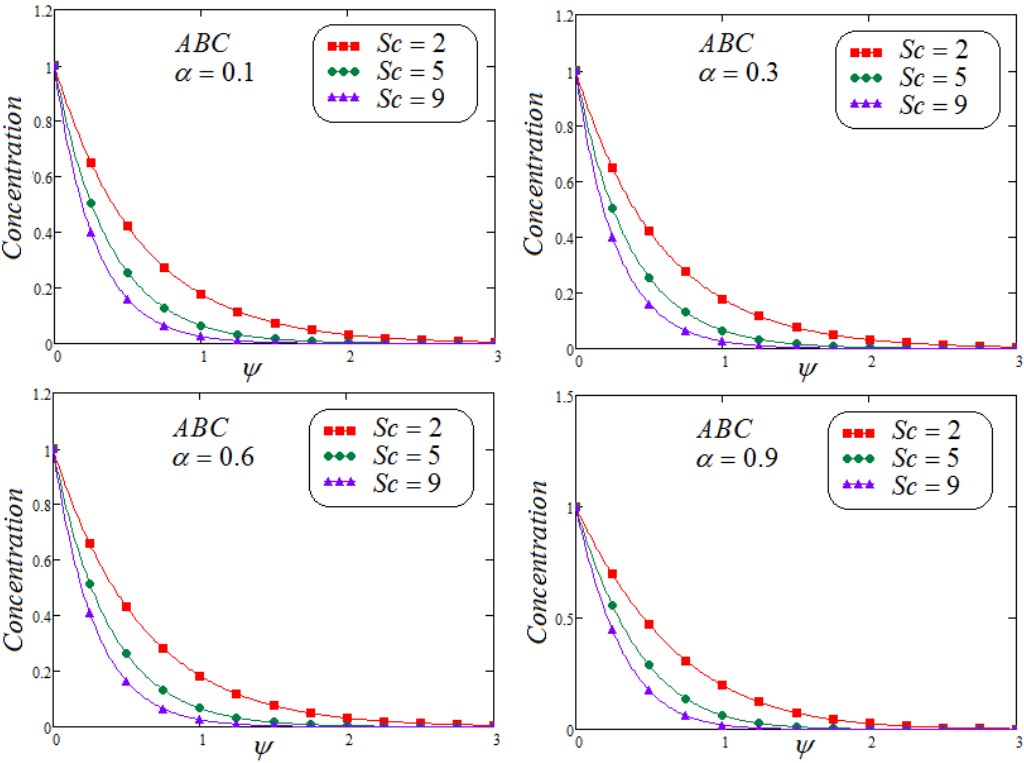

**Figure 4.** Trace of dimensionless concentration for dissimilar values of $Sc$ via ABC.

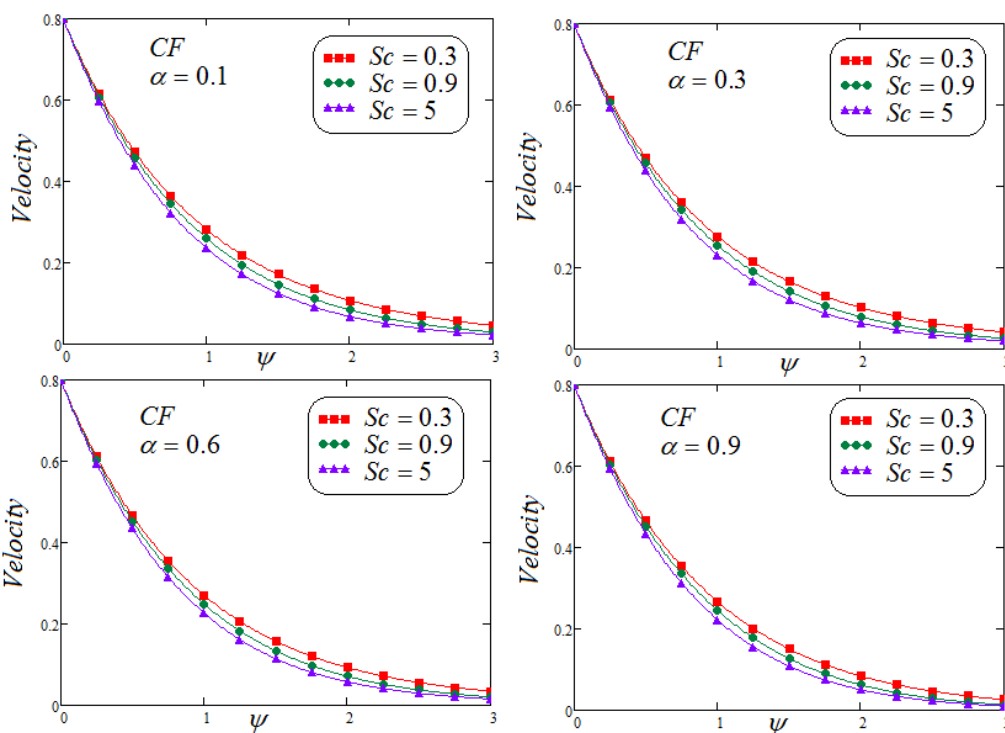

**Figure 5.** Trace of dimensionless velocity for dissimilar values of *Sc* via CF.

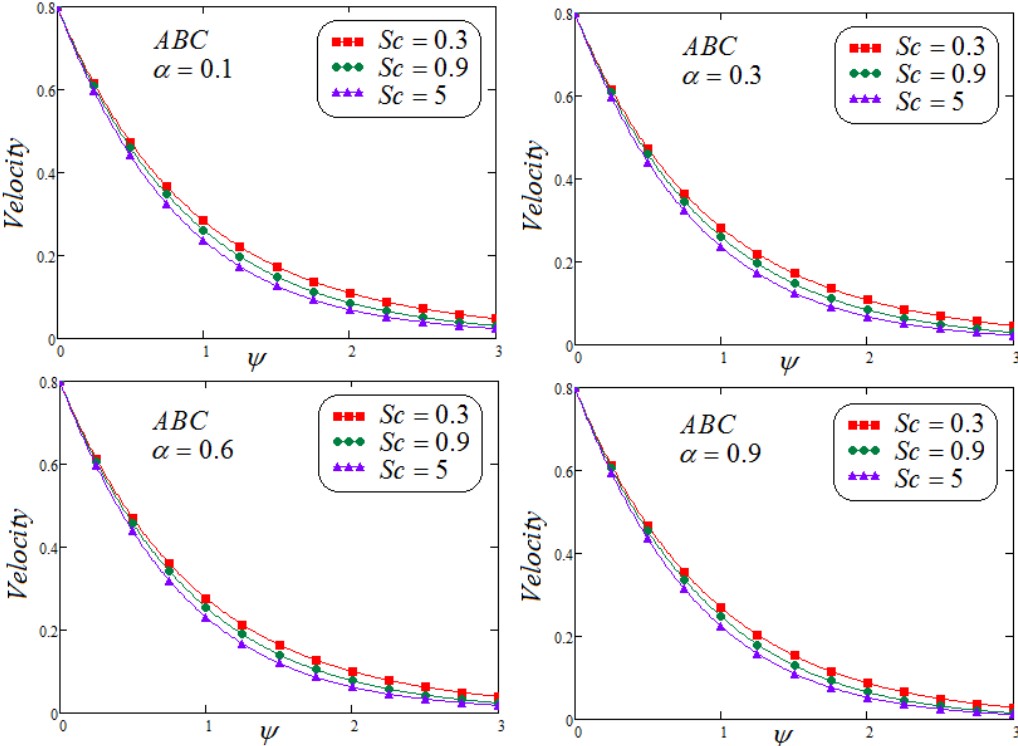

**Figure 6.** Trace of dimensionless velocity for dissimilar values of *Sc* via ABC.

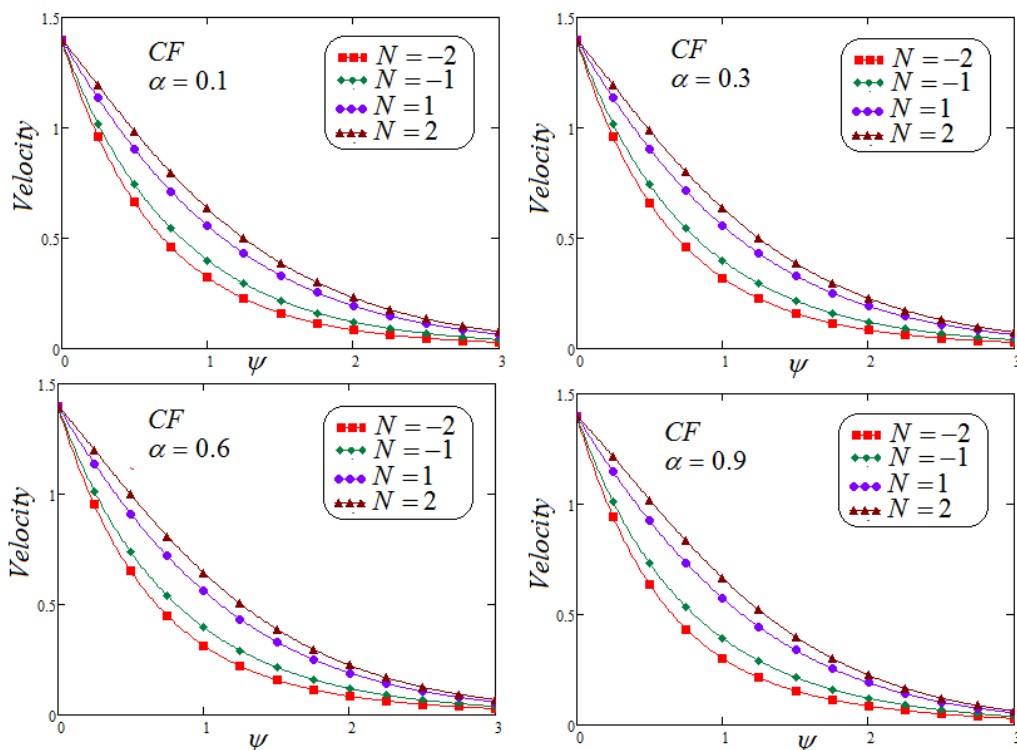

**Figure 7.** Trace of dimensionless velocity for dissimilar values of *N* via CF.

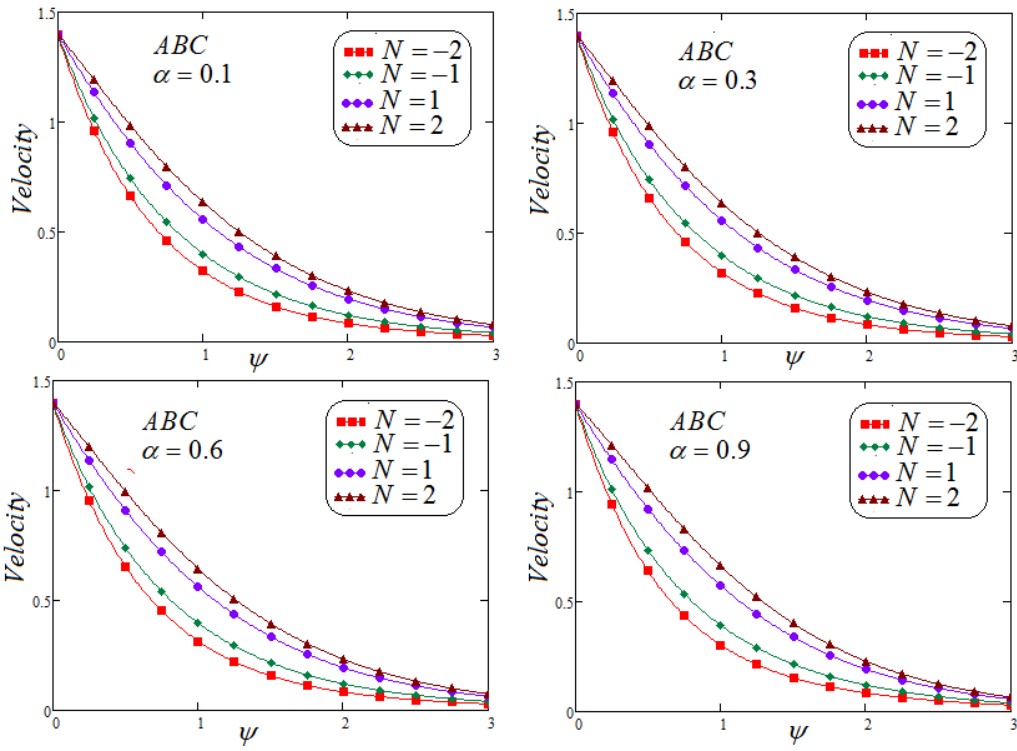

**Figure 8.** Trace of dimensionless velocity for dissimilar values of *N* via ABC.

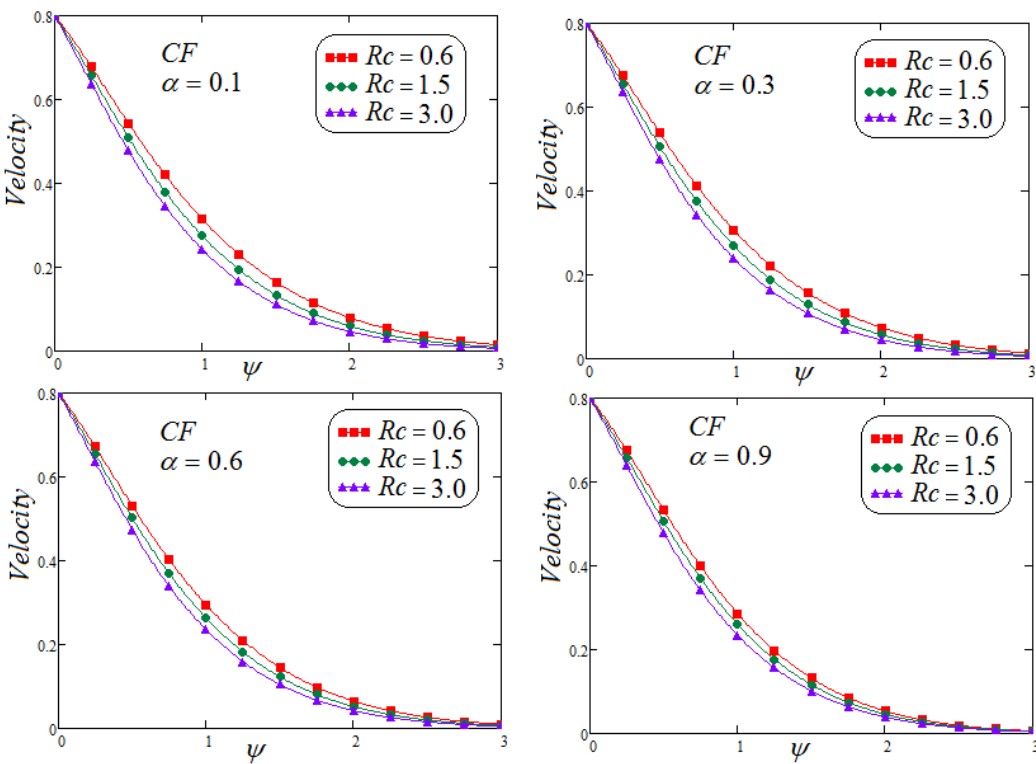

**Figure 9.** Trace of dimensionless velocity for dissimilar values of $R_c$ via CF.

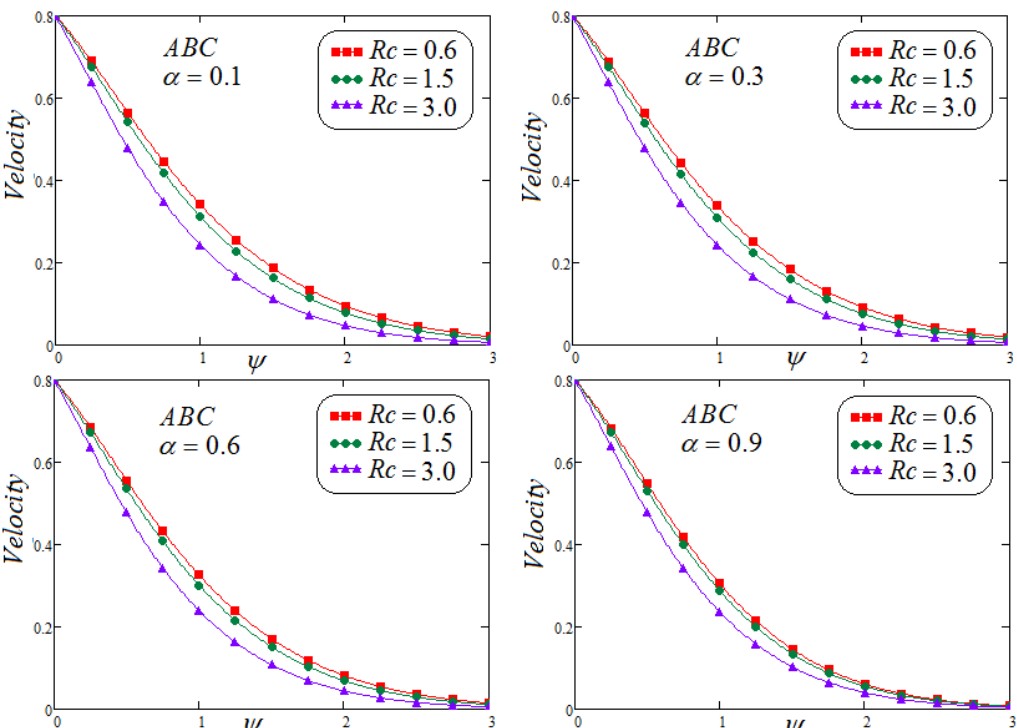

**Figure 10.** Trace of dimensionless velocity for dissimilar values of $R_c$ via ABC.

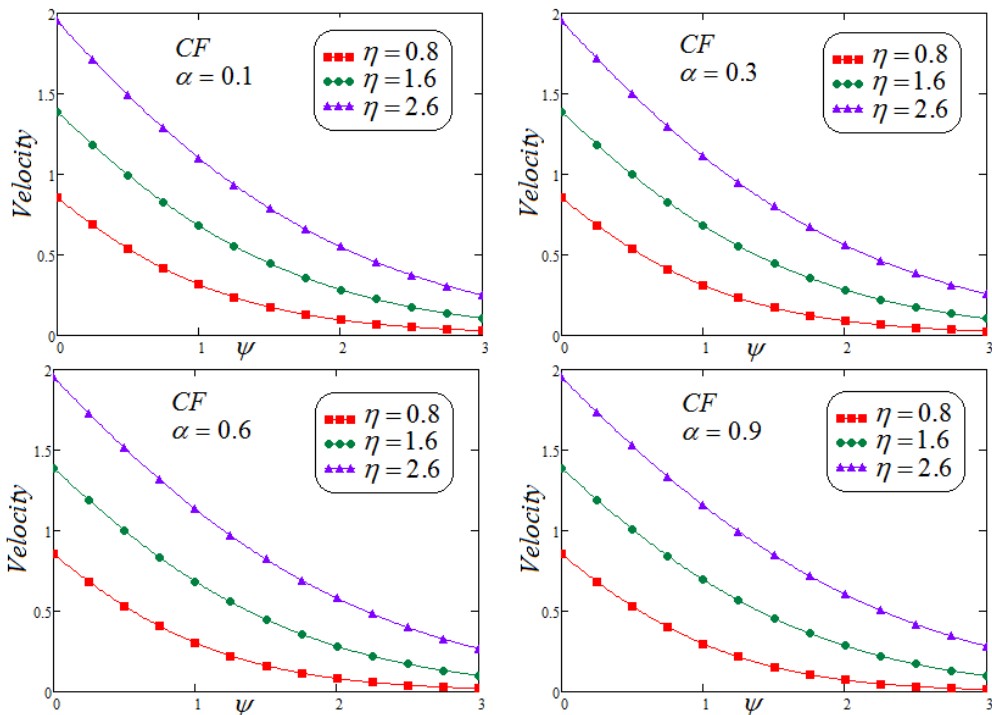

**Figure 11.** Trace of dimensionless velocity for dissimilar values of $\eta$ via CF.

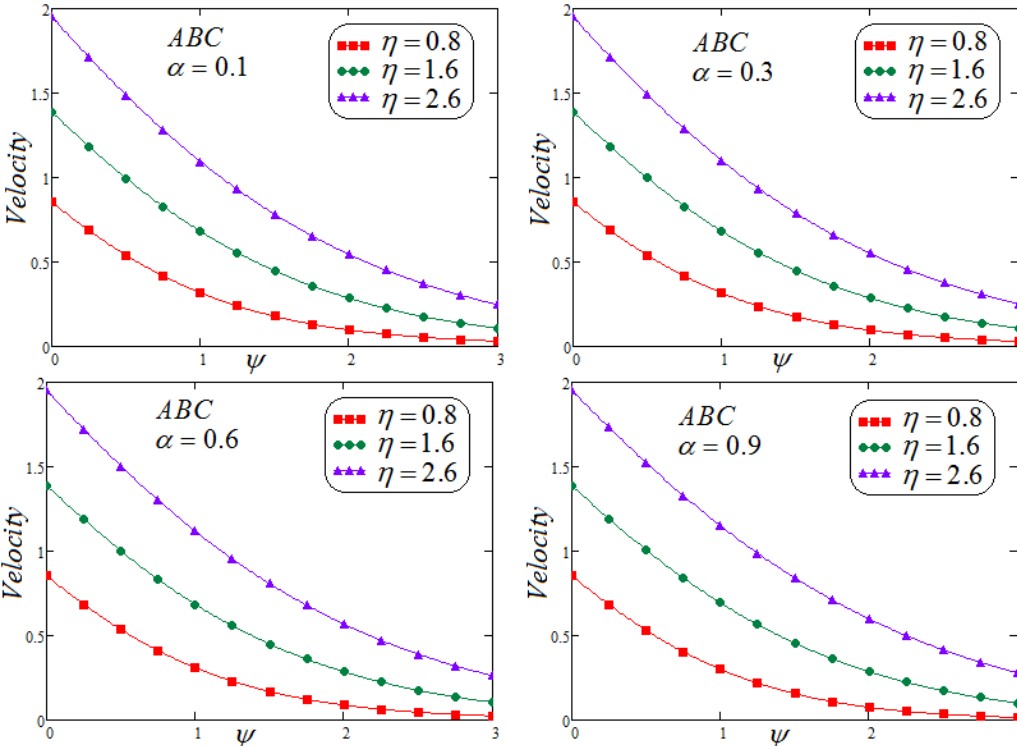

**Figure 12.** Trace of dimensionless velocity for dissimilar values of $\eta$ via ABC.

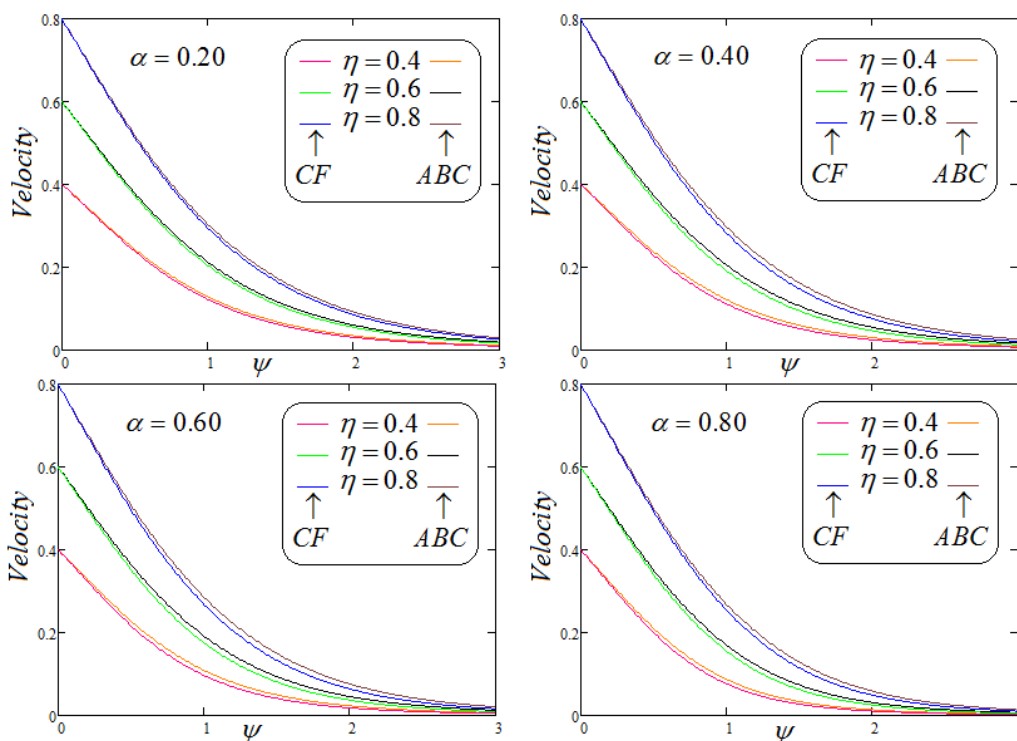

**Figure 13.** Comparison of dimensionless velocity profil for dissimilar values of $\eta$ between CF and ABC.

## 8. Conclusions

In this paper, the general equations of double diffusive magneto-free convection for viscous fluid are presented in non-dimensional form. Thermal transport phenomenon is discussed in the presence of constant concentration coupled with first order chemical reaction with exponential heating.The governing partial differential equation is inscribed into a dimensionless form. The fractional model is developed by using the modern interpretation of CF and ABC fractional time derivative operators. The Laplace transformation technique is applied to establish the analytical solution for velocity, concentration and energy equations,in terms of the generalized Lorenzo Hartly function known as the G-function for the proposed problem. Moreover, using conferred dissimilar parameters, i.e., effective Prandtl number $Pr_{eff}$, fractional parameter $\alpha$, Schmidt number $Sc$, Ratio of buoyancy forces $N$ and chemical reaction parameter $R_c$, the impacts of all these parameters on fluid velocity field, constant concentration and temperature for varying values of fractional parameter were analyzed with the help of graphical illustrations. Some noteworthy remarks and concluding results from this work are:

- It is detected that the velocity field declined with the larger values of $R_c$. Moreover, reduction in the velocity and concentration profile are observed for growing values of $Sc$ for varying values of $\alpha$.
- It is found that the fluid velocity intensifies for $N > 0$, but the opposite trend is observed for $N < 0$.
- The increasing values of the time $\eta$ stimulate the velocity distribution.
- The accumulative values of the parameter $Pr_{eff}$ decline in the temperature profile are noticed.
- Involvement of concentration factor of fluid velocity in the fluid movement is significant and cannot be overlooked.
- It is depicted that for both non-integer operators CF and ABC, velocity field, concentration and temperature profile represent the same behavior for parametric analysis of the proposed problem.

**Author Contributions:** A.U.R., M.B.R., W.R., J.A. and D.B. took part in the present research equally and significantly. All authors have read and agreed to the published version of the manuscript.

**Funding:** This work has been supported by the Polish National Science Centre under the grant OPUS 18 No. 2019/35/B/ST8/00980.

**Data Availability Statement:** During the current study no data sets were developed or investigated. So, no data sharing is applicable to this article.

**Conflicts of Interest:** The authors professed that no conflict of interest for this publication, authorship and research of this article.

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
