# Peer review of "Fractional Modeling of Viscous Fluid over a Moveable Inclined Plate Subject to Exponential Heating with Singular and Non-Singular Kernels"

_mca, doi:10.3390/mca27010008_

Round 1

Reviewer 1 Report

General comments: In this paper, the general equations of double duffusive magneto-free convection for viscous fluid are presented in non-dimensional form. Thermal transport phenomenon is discussed in the presence of constant concentration coupled with first order chemical reaction with exponential heating. The research content of this paper is very interesting, but the following questions still need to be explained.

  1. The definition of Caputo fractional operator mentioned in this paper is wrong. Please check carefully and modify later.
  2. The Symbol ℘ represents two variables in this paper, which is easy to be confused.
  3. The model in this paper is not a fractional order model. However, how to obtain the parameter α in equation (14) by Laplace Transformation technique and CF time fractional derivative? What is the value range of parameter α?
  4. In this paper, the exact solutions of the model under the definitions of CF time fractional derivative and ABC time fractional derivative are given. What is the difference between the exact solutions under the definitions of these two fractional derivatives, and whether it can be explained in this paper?
  5. What are the advantages of this method over the traditional method?

Reviewer 2 Report

In the paper, a new approach to investigate the unsteady natural convection flow of viscous fluid over a moveable inclined plate with exponentially heating is carried out. The fractionalized analytical solutions based on special functions are obtained by using Laplace transform method to tackled the non-dimensional partial differential equations for velocity, mass and energy.A numerical example is given to illustrate the effectiveness of the proposed method and the feasibility of the for the actual problem.

Fractional order differential equation is very difficult to solve the problem, sometimes even can not get solution of the problem.Solution of the Laplace transform method for fractional order equation has certain help, but in this way, most of the functions of Laplace transform and inverse transformation is very not easy to make, and sometimes do not to come out.Paper use some special functions of Laplace transform to represent the fractional order derivative, then obtains the solution of fractional order differential equation, this method has certain promotion value, published papers can be revised.

Please explain and modify the following questions:

  1. Solution of the equation is in accordance with the dimensionless fractional order differential equation, the back of the discussion in a numerical example is in accordance with the dimensionless equation or a dimensional equation.
  2. The results of a numerical example and reference (13), (15) and (38) is used in the comparison, to illustrate the effectiveness of the proposed method.The proposed method compared with integer solutions of differential equations have done?
  3. Page 5 line 126 Laplace transform under the symbol is wrong.
  4. Page 7 130 lines of the Caputoderivative definition is wrong.
  5. Please carefully check formula (15) and (24) accuracy.

Author Response

Dear Editor,

Thank you for allowing a resubmission of our manuscript in line with the reviewers’ comments and questions raised.

Reviewer#2

Comment No. 1: Solution of the equation is in accordance with the dimensionless fractional order differential equation, the back of the discussion in a numerical example is in accordance with the dimensionless equation or a dimensional equation.

Author response:  Solution of the considered problem derived from the non-dimensional governing partial differential equation.

Comment No. 2: The results of a numerical example and reference (13), (15) and (38) is used in the comparison, to illustrate the effectiveness of the proposed method. The proposed method compared with integer solutions of differential equations have done?

Author response:  Done as per suggestions.

Comment No. 3: Page 5 line 126 Laplace transform under the symbol is wrong.

Author response:  The suggested correction has been made.

Comment No. 4: Page 7 130 lines of the Caputo derivative definition is wrong.

Author response:  Done as per suggestions.

Comment No. 5: Please carefully check formula (15) and (24) accuracy.

Author response:  The suggested correction has been made.

Reviewer 3 Report

In this paper, the authors have studied unsteady natural convection flows of viscous fluids over a moveable inclined plate with exponentially heating. The authors considered a generalized mathematical model based on fractional differential equations.

The closed forms of solutions of the proposed problem have been determined using Laplace transform. The influence of the memory parameter on fluid motion and heat transfer is investigated by numerical simulations and graphical illustrations.

The paper needs a serious review because the present form contains many errors.

Observations:

1). The information regarding the Rosseland approximation are not complete. The authors must add some relations linked with Eq. (7) to obtain the radiation parameter N_r and the form (9) of the energy equation (See, for example, the reference [11]).

2). The nondimensional temperature “theta” is incorrectly defined. Must be “theta” = (T-T_inf) / T_w. If at the denominator is T_inf, you cannot obtain the boundary condition “theta”(0, eta) = 1- a exp(-b eta).

3). Verify and correct equations (6) and (13). Instead of “Inf” must be T_inf, respectively, instead of “Inf” must be zero.

4). The Laplace transform of the two-parameter Mittag-Leffler function is wrong written. At denominator must be “Gamma(p*l+beta)”.

5). The definition of the Caputo fractional derivative is completely wrong. In this manuscript the authors used two parameters "m" and "n")!!! The condition for the fractional parameter "p" is wrong.

6). The definition of Caputo-Fabrizio derivative is wrong. For the exponential function must be “eta-tau” instead of “eta-p”.

7). The definition of the Atangana-Baleanu fractional derivative is wrong.

8). The expression “theta_1” is wrong. Verify and correct the inverse Laplace transform. The argument of the exponential function is incorrect.

9). In all figures, the authors used the variable "y" instead of the variable "psi" that is used in the text. It is confusing.

10). It would be good if the authors presented some graphs for the fluid temperature. Are too many graphs for velocity and none for temperature and concentration.

Round 2

Reviewer 3 Report

The manuscript contains errors. A revision is needed.

1). Verify and correct the definition of Caputo derivative. There is no agreement between the arguments of the function f on the left-hand side, respectively right-hand side.

2). Verify and correct the definition of Atangana-Baleanu derivative. At the Mittag-Leffler kernel, the argument (eta-tau) is at power 1. This is not correct. The power must be equal to the index of the kernel.

3). The boundary condition for the dimensionless concentration is C(0,eta) = 1(See Eq. (12)). In figures 3 and 4 this condition is unsatisfied. In these figures C(0, eta) = 0.8. Therefore, results regarding the concentration are incorrect.

Author Response

Reviewer#3

Comment No. 1: Verify and correct the definition of Caputo derivative. There is no agreement between the arguments of the function f on the left-hand side, respectively right-hand side.

Author response:  Done, definition corrected as per suggestion.

Comment No. 2: Verify and correct the definition of Atangana-Baleanu derivative. At the Mittag-Leffler kernel, the argument (eta-tau) is at power 1. This is not correct. The power must be equal to the index of the kernel.

Author response:  Done, removed the typos in the definition as per suggestion.

Comment No. 3: The boundary condition for the dimensionless concentration is C(0,eta) = 1(See Eq. (12)). In figures 3 and 4 this condition is unsatisfied. In these figures C(0, eta) = 0.8. Therefore, results regarding the concentration are incorrect.

Author response: Done, removed the minor error, so the figures 3 and 4 satisfied the boundary condition of concentration. All equations are correct and checked carefully.

Round 3

Reviewer 3 Report

In the revised form, the authors made the requested corrections.